# Effects of High-Intensity Interval Training and Continuous Aerobic Training on Health-Fitness, Health Related Quality of Life, and Psychological Measures in College-Aged Smokers

**DOI:** 10.3390/ijerph20010653

**Published:** 2022-12-30

**Authors:** Nduduzo Msizi Shandu, Musa Lewis Mathunjwa, Brandon Stuwart Shaw, Ina Shaw

**Affiliations:** 1Department of Human Movement Science, Faculty of Science and Agriculture, University of Zululand, KwaDlangezwa 3886, South Africa; 2School of Sport, Rehabilitation and Exercise Science, University of Essex, Colchester CO4 3SQ, UK

**Keywords:** smoking, exercise, pulmonary function, cardiovascular health

## Abstract

The study examined the effects of exercise on health-fitness, health related quality of life (HRQOL), and psychological measures in college-aged smokers. Outcomes included HRQOL, hemodynamic, anthropometric, lung function, and cardiorespiratory endurance. Sixty physically inactive college-aged male smokers (18–30 years) were randomly assigned into three groups: high-intensity interval training (HIIT), continuous aerobic training (CAT), and a control (CON). Both HIIT and CAT groups completed 8 weeks of non-consecutive cycling sessions thrice weekly. The CON group were not subjected to the exercise intervention. Sixty participants met the inclusion criteria. Of these, 48 (HIIT: *n* = 18, CAT: *n* = 16, CON: *n* = 14) participants completed the study and were included in the final analysis. Compared to CON, HIIT significantly (*p* = 0.01) improved forced expiratory flow (FEF_75%) more than the CAT group (*p* = 0.29). HIIT provided a significant (*p* = 0.04) improvement in FEF_75% compared to CAT. Recovery heart rate (RHR) was significantly improved in participants assigned to HIIT (*p* = 0.00) and CAT (*p* = 0.002) groups compared with the CON. A significant difference in RHR was found in HIIT compared to CAT. The study findings indicate that both HIIT and CAT exercise interventions significantly improve markers of lung function and cardiorespiratory endurance, respectively. However, findings suggested that HIIT should be the preferred form of exercise regime among college-aged smokers for more significant, healthier benefits.

## 1. Introduction

Smoking is a major health problem and the cause of most premature mortalities, with increasing rates in many developing countries worldwide [1]. This is supported by the worldwide social developments and greater rationalisations of tobacco market transactions in these countries [1]. The World Health Organization (WHO) has estimated that the mortality rate caused by tobacco smoking will rise over eight million in the following decades and account for six million deaths annually, particularly in developing countries [2]. Moreover, according to WHO estimates, around 47% of men and 12% of women smoke cigarettes/tobacco worldwide [3]. While developed countries account for 42% of men who smoke, compared to 24% of women, in developing countries, 48% of men and 7% of women smoke [1]. Gender statistics of smokers in South Africa support WHO estimates, with 32% of men and over 8% of women already smoking tobacco [3].

The high concentrations of nicotine have been associated with causing physical and psychological dependency with prolonged tobacco smoking trends [4]. Several studies have demonstrated significant declines in the HRQOL of smokers compared to non-smokers, irrespectively: physical health, psychological state, social relationships, and environmental points [4,5,6,7,8]. Apart from other numerous complex psychological risk factors, reports have noted significant evidence of the detrimental effects that smoking exerts on the human body, such as the development of coronary artery disease, cardiovascular disease, peripheral vascular disease, myocardial infarction, stroke, and sudden death [7,9].

Engagement in exercise has been noted to reduce depressive indications, widely reduce depressive indications and mental health issues, and reduce the risks of the burden of cardio-metabolic diseases, not only among compromised and/or depressed patients but also in apparently healthy individuals [10,11,12,13]. However, exercise intensity, duration, and frequency define the magnitude of responses to health fitness, cardiovascular endurance, HRQOL, and mental well-being [5,8]. Continuous aerobic training has been considered vital for improving lipid profile and cardiovascular fitness for all populations, both patients and healthy individuals [14,15]. While HIIT is a type of exercise that involves short, sharp bursts of activity, followed by periods of active rest. [6]. Greater or superior improvements in cardiometabolic fitness can be attained using HIIT as it can be adapted to a variety of physical fitness levels [10]. Therefore, it is well established that HIIT may provide a solution, given that the total amount of time spent per week is substantially less than that of traditional aerobic exercise to accommodate their studies [1]. Moreover, HIIT alternates short periods of intense activity with less intense recovery periods of light exercise, is more engaging and enjoyable than traditional continuous aerobic (CA) exercise interventions, and has greater physiological benefits [15,16,17]. This study aims to determine the effectiveness of high-intensity interval training (HIIT) and continuous aerobic training (CAT) on health fitness, health-related quality of life (HRQOL), and psychological measures of college-aged smokers.

## 2. Materials and Methods

### 2.1. Participants

This research was approved by the Institutional Review Boards of the University of Zululand (Reg No: 171110-030), Republic of South Africa. A total of 60 male smokers aged 18–30 years were recruited through advertisements at the University of Zululand, KwaDlangezwa Campus.

### 2.2. Study Population and Sample

This study was a randomised controlled trial (RCT) consisting of two intervention groups, HIIT (*n* = 20) and CAT (*n*= 20), and one control group, CON (*n*= 20). This particular cohort was chosen purposefully for the study, since men are typically associated with 4–5 times increased smoking rate and exposure to premature mortality risk than women, which is often depicted in middle-aged individuals. The recruitment strategy only mentioned that the study was an 8-week voluntary intervention for improving quality of life and physical fitness in middle-aged male smokers, without specifying the exercise protocols (i.e., to further prevent selection bias). Participants were considered smokers if they currently smoked or quit within six months from the assessment, and physically inactive if they reported less than 150 min of moderate or 75 min of vigorous PA per week, as assessed via the International Physical Activity Questionnaire (IPAQ) [16]. Exclusion criteria included (1) smokers with high blood pressure (>180/100 mm Hg); (2) taking prescribed medication for chronic health and medical conditions, including but not limited to myocardial infarction, uncompensated heart failure, or unstable angina pectoris; and (3) any pre-existing medical or physical issue that could affect training and experimental tests, as outlined by current exercise prescription guidelines. All participants were required to be free of any absolute or relative contraindications to exercise [16]. Participants provided written informed consent/signed the consent form to participate in the study, and all participants were informed of their right to discontinue the survey at any point.

### 2.3. Assessment

#### 2.3.1. Health-Related Quality of Life (HRQOL) Assessment

The World Health Organization Quality of Life (WHOQOL-BREF) questionnaire was used to measure each participant’s perceived HROQL before and following the 8-week exercise intervention period. The questionnaire targeted mainly physical health, social relationships, psychological state, and environmental points. Thus far, several studies have analysed and confirmed the accuracy and precision of high internal validity with Cronbach’s α coefficient of 0.81 [4,7,8].

#### 2.3.2. Hemodynamic Assessment

Blood pressure was measured according to the standards established by the American College of Sports Medicine [16] using a sphygmomanometer and stethoscope (Jiangsu Dengguan Medical Treatment Instrument Co. Ltd., Changzhou, China), after a five-minute rest using the auscultatory method. A polar heart rate monitor (Polar F4M BLK, Finland) was used to measure resting HR after a five-minute rest [17]. The rate pressure product (RPP) was calculated by the following formula: RPP = (SBP × HR) × 10^−2^ [18]. Mean arterial pressure (MAP) was assessed using the following formula: MAP = SBP + 2 (DBP)]/3 [19].

#### 2.3.3. Anthropometric Assessment

All anthropometric measurements were performed by a trained biokineticist according to the methods proposed by the International Society for the Advancement of Kinanthropometry (ISAK) [20]. Height and body weight were assessed with the participant in lightweight clothing and shoes removed. Body weight (kg) was assessed using a calibrated weight scale platform to the nearest 0.1 kg, and participants stood barefooted in the centre (Detecto, Mediotronics (PTY) LTD, Durban, South Africa). Stature was measured using a standardised wall-mounted stadiometer (Seca Stadiometer, 216, Seca, Birmingham, UK). Body mass index (BMI) was calculated from height and weight measurements. Skinfolds (triceps, subscapular, suprailliac, abdominal, thigh, and calf) were taken on the right side of the body using a Harpenden skinfold calliper (Harpenden, HSB-BI, ATICO Medical Pvt. Ltd., UK), and percentage body fat (%BF) was calculated using the equation of Jackson and Pollock (1978): percentage fat = 100 (4.95/body density (Db) − 4.5), where Db (g/cc) = 1.120 − 0.00043499 (sum of the seven skinfolds in millimetres (∑7)) + 0.00000056 (∑7) − 0.00028826 (age)) [21,22]. Waist and hip circumferences (as utilised in the waist-to-hip ratio (WHR)) were measured using a non-distendable measuring tape (Holtain Ltd., Crymych, UK) and WHR was calculated by the following equation: WHR = waist circumference ÷ hip circumference [19]. Fat mass was calculated by multiplying body mass by body fat percentage divided by 100. Lean mass was calculated as total body mass in kilograms subtracted by fat mass in kilograms.

#### 2.3.4. Lung Function Assessment

A calibrated electronic spirometer (Chest Graph, HI-101, CHEST LTD., Tokyo, Japan) with a precision volume of ±50 mL or ±5% was used. Participants were required not to smoke or eat within one hour prior to testing. The technique involved testing participants seated with nose clips to ensure isolated breathing through the mouth. The highest value recorded from three trials of forced vital capacity (FVC) manoeuvre was the included value in the final analyses for each participant. Spirometer measures included: FVC, forced expiratory volume in one second (FEV_1_), FVC/FEV_1_ ratio, peak expiratory flow (PEF), and forced expiratory flow after 25%, 50%, and 75% of vital capacity has been expelled (FEF_25_, FEF_50_, and FEF_75_, respectively); these were utilised in the present study.

#### 2.3.5. Cardiorespiratory (CR) Endurance Assessment

A one-minute step test and standardised metronome (30 beeps per minute) were used to assess cardiorespiratory (CR) endurance, measured as VO_2max_ [23]. Heart rate response was utilised to calculate VO_2max_ and measured in mL^−1^·kg^−1^·min^−1^ and was calculated using the following equation: 3.5 + 0.2 × steps·min^−1^ + 1.33 × (1.8 × step height (m) × steps·min^−1^) [23].

#### 2.3.6. Intervention

All participants in the CON group were required to remain sedentary for the duration of the eight-week experimental period. In contrast, the CAT and HITT participants were required to participate in their assigned thrice-weekly, non-consecutive exercise sessions for eight weeks. The HIIT intervention consisted of three 33 min sessions of non-consecutive exercise per week [24]. This programme was completed on a stationary cycle ergometer (Mornark 874E, Vansbro, Sweden) at the University of Zululand gymnasium (Marius Coetzee gym) with a qualified biokineticist. Participants began with a five-minute warm-up at an intensity corresponding to 65–75% HR, followed by eight seconds of cycle sprinting, and twelve seconds of passive rest for a maximum of 60 repetitions on a cycle ergometer. The initial resistance was 1.0 kg and participants exerted as much effort as possible during the sprinting phase. Since participants were physically inactive individuals who also smoked, we utilised a progressive programme to account for adaptation, with a biokineticist continually increasing the resistance by 0.5 kg if a participant completed two consecutive 20 min intermittent sprinting exercise sessions. After the main exercise, participants conducted a cool-down session through cycling for 5 min at 20–40% HR_max_ followed by standard stretches of the calf, hamstring, quadriceps, gluteus, back, neck, and shoulder muscles held for 20 s at an intensity whereby the participant experienced no pain in the position [25,26].

The CAT exercise intervention included three 55 min non-consecutive sessions per week, conducted in the presence of a biokineticist. Both the warm-up and cool-down were performed for five minutes at 20–40% of the maximum heart rate (HR) [18,20]. The following equation was used to calculate HR to obtain the targeted training zones: HR = 220-age [19]. Participants performed continuous cycling at 60–75% of VO_2max_ on a cycle ergometer (Mornark 874E, Vansbro, Sweden) for 40 min [18], and maintained a cycling speed of 60 ± 5 rpm throughout each training session. To account for adaptation, cycling resistance was increased for participants who adapted to the training programme to maintain the intensity at 60–75% of maximal oxygen consumption (VO_2max_). Each session concluded with stretching the calf, hamstring, quadriceps, gluteus, back, neck, and shoulder muscles [25,26]. For the stretching exercises, participants completed one set of each flexibility exercise held for 20 s at an intensity whereby the participant experienced no pain in the position [25,26].

#### 2.3.7. Statistical Analysis

The study used quantitative research methods, involving already established physical testing assessments. The proposed study used the statistical analysis programme Statistical Package for Social Sciences (SPSS) version 22 for Windows (SPSS Inc., Chicago, IL, USA), which calculated the descriptive statistics, including the means and standard deviations of the data collected. Paired-samples *t*-tests were utilised to examine the differences between pre-test and post-test variables. The effect size was calculated using the statistical calculation d Cohen (1988), and the standardised effect sizes were classified as small (<0.20), moderate (0.20 to 0.79), and large (>0.80). Data were also processed using analysis of variance (ANOVA), regression analysis, and a subsequent independent *t*-test. Moreover, the relationship between the continuous data was assessed using Pearson’s correlation coefficient.

## 3. Results

Of the initial 60 students who were eligible to participate in the study, 48 male student smokers (HIIT: *n* = 18, CAT: *n* = 16, Con: *n* = 14) completed the survey and were included in the final analysis. Sixteen participants were excluded from analysis in the study as they were unable to complete the eight-week experimental period and be assessed at post-testing. Participants in the HIIT group were more adherent to the prescribed exercise intervention compared to the participants in the CAT group. There were no other statistically significant group differences at baseline.

Following the 8-week intervention, both HIIT and CAT groups indicated more significant improvements in the HRQOL domains (*p* < 0.05) than the control group (*p* > 0.05) (Table 1). However, the social relationships score of the CAT group indicated otherwise—raw data (3–15): 12.25 ± 1.48, *p* = 0.142; transformed score (4–20): 16.38 ± 2.06, *p* = 0.177; transformed score (0–100): 77.38 ± 12.96, *p* = 0.181) at post-intervention. Interestingly, a positive change was noted among the control group in the environmental domain score (raw data: 45.00 ± 14.45, *p* < 0.003; transformed score: 11.29 ± 2.13, *p* < 0.000; transformed score: 45.00 ± 14.45, *p* < 0.003). For hemodynamic variables, only HIIT and CAT significantly improved RHR (66.61 ± 9.15, *p* = 0.009 and 64.13 ± 5.16, *p* = 0.000, respectively) and RPP (66.64 ± 10.93, *p* = 0.042 and 71.57 ± 8.12, *p* = 0.000, respectively) from baseline (Table 2).

The results for anthropometric variables indicated that both exercise programmes had different effects on the components of body composition after 8 weeks (Table 3). Following the intervention, in the HIIT and CON groups, a significant (*p* < 0.05) difference was shown in weight (67.90 ± 7.61, 1.8 increase and 70.77 ± 6.04, 3.2 decrease, respectively) and BMI (22.98 ± 2.10, 1.8 increase; 23.65 ± 2.61, 3.7 decrease, respectively). Nonetheless, the exercise intervention performed in the study had different intensities and duration, a significant (*p* < 0.05) difference was found across all groups in the sum of skinfolds, particularly improvements occurred in HIIT (73.61 ± 15.95; 15.7 increase) and CAT (88.56 ± 17.10; 16.1 increase), while reductions were seen in the control group (61.71 ± 9.36, *p* = 0.005) post intervention. Moreover, the results show that HIIT and CAT groups indicated significant (*p* < 0.05) improvements in %BF from 8.06 ± 288 to 10.54 ± 3.20—a 23.5 increase, and from 10.03 ± 2.19 to 12.23 ± 2.58—a 18.0 increase, respectively, and fat mass from 5.39 ± 2.11 to 7.19 ± 2.50—a 25.0 increase, and from 7.51 ± 2.62 to 9.15 ± 2.93—a 17.9 increase, respectively. Strikingly, lean mass indicated significant (*p* < 0.05) reductions in the CAT group from 66.25 ± 10.34 to 64. 34 ± 8.53—a 3.0 decrease, and from 66.42 ± 4.67 to 64.52 ± 5.50 in the Con group.

The study indicated that the exercise intervention targeted lung function components differently in each group and statistical significance was mostly evident in HIIT, followed by control, as compared to CAT (Table 4). HIIT showed improvement in FVC–meas (4.36 ± 0.69, mean ± SD); *p* < 0.05), FEV–Pred% (93.78 ± 9.66, mean ± SD; *p* < 0.05), FEF_50–Meas (4.60 ± 0.51, mean ± SD; *p* < 0.05). FEF_50–Pred% was found to be significant in both HIIT and control groups by 85.22 ± 7.43, mean ± SD and 76.49 ± 7.06, mean ± SD; *p* < 0.05, respectively, from baseline. CAT indicated significant improvements in FVC–Pred% (86.76 ± 11.49, mean ± SD; *p* < 0.05) and FEV–meas (3.81 ± 0.35, mean ± SD; *p* < 0.05), while baseline PEF in CAT and control, both measured and predicted, was indicated to be significant among smokers by 9.30 ± 1.28 and 7.06 ± 1.34, mean ± SD; *p* < 0.05 and 97.04 ± 10.72 and 79.30 ± 12.79, mean ± SD; *p* < 0.05, respectively. Both measured and predicted FVC/FEV_1_ (%) were only significant in the control group (*p* < 0.05), as well as FEF_75–Pred% (*p* < 0.05).

In all three groups—HIIT, CAT, and control—no significant differences were noted in VO_2max_ (27.47 ± 1.00, mean ± SD, *p* = 0.123; 26.98 ± 2.12, mean ± SD, *p* = 0.226 and 24.05 ± 2.34, mean ± SD, *p* = 0.437, respectively) for the cardiorespiratory endurance of smokers (Table 5). However, a statistically significant difference was noted in the heart rate of CAT members (86.19 ± 12.17, mean ± SD; *p* < 0.05).

## 4. Discussion

This study aimed to determine which exercise type, between high-intensity interval training and continuous aerobic training, provides greater significant benefits to health fitness, health-related quality of life, and psychological measures in college-aged smokers. Further, the study aimed to identify the effective strategy to curb the risk of premature mortality caused by smoking-attributable diseases. The findings may be substantial for healthcare professionals to administer accurate and optimal treatment and management regimes for smokers. Previous studies indicate that nearly one-third of students in university resort to smoking as they transition in life (studies, assignments, fees, etc.) [2,6,9].

Engagement in exercise improves the physiological and psychological processes of human functioning [15,24]. Moreover, a strong relationship between exercise and health fitness, mental well-being, and HRQOL has been reported in the literature [24]. Interestingly, our findings mirror those of the previous studies that have examined HIIT and CAT to significantly improve HRQOL measures in smokers [8]. One unexpected finding was the extent to which social relationships in smokers were only improved considerably in the HIIT group (*p* < 0.05). A possible explanation for this might be that participants benefitted from short workout periods, and they built on conversations more through socializing [15]. Another important finding was the significant difference seen in the environmental domain across all groups. These results corroborate the ideas of Craig et al. (2009), who suggested that exercise positively changes the environmental status or lifestyle of a smoker, such as with improved sanitary measures and/or precautions [27]. It is worth to note that, at present, current studies have found no significant difference in HRQOL between smokers and non-smokers [26,27]; our study exercise intervention can assist to curb this gap in knowledge, as it has successfully identified that significant differences exist in the HRQOL of smokers who do and do not exercise, with greater benefits specifically coming from high-intensity interval training.

Our hemodynamic results confirm that exercise alters the cardiac sympathetic–parasympathetic balance with significant changes observed in RHR and RPP of smokers in HIIT and CAT groups following the intervention (*p* < 0.05). The study indicated a decreased percentage of change of RHR from 5.5, ES = 0.667 to 9.4, and ES = 1.424, respectively, and RPP from 10.1, ES = 0.495 to 10.8, and ES = 1.091, respectively. In accordance with the present results, previous studies have demonstrated that RHR and RPP exhibit a dose-dependent response to the workload to adequately provide blood during exercise to the active myocardium [28,29]. This result may be explained by the fact that exercise stimulates the autonomic control centre to increase the cardiac sympathetic activity and decrease the effectiveness of the cardiac parasympathetic nervous system, which results in increased HR, stroke volume, and cardiac output, and assists in redistributing blood flow to the active skeletal muscles [7,9]. However, this study could not demonstrate a significant relationship between systolic blood pressure, diastolic blood pressure, and mean arterial pressure (*p* > 0.05) across all groups. These results reflect those of Kim et al. (2012), who also found no significant changes in pre-test to post-test effects of SBP from 119.5 ± 8.9 to 117.8 ± 10.3 and 118.8 ± 10.9 to 124.8 ± 11.3 for exercise smoker and non-exercise smoker groups, respectively, with *p* > 0.05 for both groups [30]. It is difficult to explain this result, but it might be related to the study design and population. Another possible alternative explanation of our findings is that it could be conceivably hypothesised that smokers may require more extended engagement in exercise, as smoking has been identified to alter skeletal muscle fibres and reduce the capability of oxidative enzymes, which results in skeletal muscle dysfunction [6,29].

Most studies have indicated that the body composition of smokers in college changes during the freshman/sophomore year or the complete college years with weight gain being attributed to increases in fat mass [30,31,32]; although, some reports indicate weight gain without changes in fat mass [31,32]. Efendi et al. (2018) showed that in smokers, the overall reduction in total body mass is attributed to the significant decrease in the amount of fat mass, total body, and visceral fat that occurs during aerobic exercise training [4]. Nevertheless, studies have indicated that greater benefits can be obtained earlier with HIIT, as significant changes in smokers are associated with the duration and intensity of training [33,34]. Increases in fat oxidation during training have been deemed the cause [34]. Our results are in accordance with recent studies indicating that HIIT extensively improves weight gain, BMI, the sum of skinfolds, percentage of body fat, and fat mass compared to CAT [26]. Interestingly, there was a significant difference in the BMI, the sum of skinfolds, and the lean mass of individuals in the control group. According to these data, we can infer that our study findings may highlight some previous findings that exercise alters body composition in smokers, providing specificity on the mode, duration, and frequency of exercise training. This can be useful, in order to acquire more significant, healthier benefits over a short period during treatments/rehabilitation and intervention sessions in private and public health [31,32].

The most obvious finding to emerge from the analysis is that after an 8-week exercise intervention period, most smokers’ lung function parameters improved significantly across all groups. Although consistent with the literature, this research found that some participants’ lung function did not improve from baseline [4]. Perhaps the most striking finding is that improvements in FVC and FEV occurred only in intervention groups. This finding broadly supports the work of other studies in this area linking exercise with lung function improvements in smokers [4,35]. These results support the hypothesis that measured FVC and predicted FEV improve with high-intensity exercise, and predicted FVC and measured FEV improve with a long duration type of training, in smokers. However, more research on this topic must be undertaken before the association between exercise and measured/predicted lung function parameters is more clearly understood. This finding is contrary to previous studies which have suggested that no significant difference exists in FVC/FEV_1_ for both HIIT and CAT groups [36]. One unanticipated result was that the control group indicated a significant reduction in the ratio (*p* > 0.05). Moreover, FEF values were mostly improved and reduced in the HIIT and the control groups, respectively. This finding is contrary to previous studies, which have suggested no relationship between exercise and FEF in smokers [30,31,37]. These conflicting experimental results could be associated with the nature of the environment, sample size, population, gender, and assessment, as well as the intervention protocols utilised.

It is generally accepted that people who report higher levels of exercise tend to have higher levels of fitness, and exercise can improve cardiorespiratory fitness [37], although studies among smokers report otherwise [34]. In contrast with the literature, our study shows no statistical significance in VO_2max_ as a cardiorespiratory indicator across all groups. Gellert et al.’s (2015) 12-week training programme significantly improved the VO_2max_, HR, systolic BP, and diastolic BP, and was similar to Jonas et al. (1992) who used the same protocol to assess participants’ physical capacity [38,39]. However, interestingly, the only evidence of HR improvement in our study was detected specifically in CAT. A possible explanation for these results may be the inadequate monitoring of equipment during the intervention period. Therefore, a further study focusing on changes in cardiorespiratory endurance of smokers from baseline, during, and following the intervention period is suggested.

The present study confirms previous findings and contributes additional evidence that suggests that training programmes of such durations have major effects on overall quality of life, mainly psychological components, hemodynamic processes, spirometry, and cardiorespiratory variables, with extensive positive outcomes in HIIT. Additionally, these findings contribute in several ways to our understanding of the benefits of exercise and its effects on cardiovascular and pulmonary function in smokers, specifically in the duration and frequency of exercise. This study set out to critically examine the effects of HIIT and CAT on health-fitness, HRQOL, and psychological measures in college-aged smokers. Returning to the question posed at the beginning of this study, it is now possible to state that while several benefits have been previously demonstrated from engagement in exercise [35], our study suggests that high-intensity interval exercise training may be useful in improving health and mental well-being, slowing the risks of onset and progression of smoking-attributable diseases and indirectly, prolonging life expectancy in smokers [36].

### Limitations and Directions for Further Research

The strengths and limitations of the study must be acknowledged. While this experimental study was strengthened by using a randomised study design and collection of data via validated measures, there were limitations. First, the study consisted of a small, highly educated, homogenous sample which limits the generalisability of our results, and the power to detect statistically significant differences. Second, the findings may not be generalisable to other age groups, given the emphasis on smoking among adults aged 18–30 years. However, the results of the findings are consistent with previous evidence on the relationship between exercise and health fitness, HRQOL, and psychological measures. Additionally, the randomisation groups were not exchangeable (e.g., unequal participants at post-test); therefore, our observations may be confounded. To significantly address the limitations, upcoming studies should be structured and conducted with a larger and more diverse study sample. Third, the psychometric analysis of the satisfaction survey was not conducted, and therefore, the validity and reliability were not determined.

## 5. Conclusions

In summary, the high-intensity interval training programme used in the present study significantly improves health fitness, HRQOL parameters, and psychological measures in college-aged smokers, as compared to continuous aerobic training. The conclusions are consistent with the results of several published studies, which indicated that exercise is a vital component in decreasing the related risks of smoking-attributable diseases and can possibly increase life expectancy, which may be associated with physiological and psychological benefits [4,37,39]. Because the exercise intensity was scientifically adapted to participant capacity, we may suggest that both HIIT and CAT programmes are suitable for use from public health and clinical points of view when dealing with smokers.

## Figures and Tables

**Table 1 ijerph-20-00653-t001:** Intra-group comparisons of health-related quality of life variables (raw data and transformed scores): physical, psychological, social relationships, and environment in college-aged smokers following 8 weeks of high-intensity interval training (HIIT), continuous aerobic training (CAT) and a control group (Con).

Variable	Group	Baseline	8-Weeks	%∆	Effect Size: Cohen’s d	*p*-Value
Physical State						
Raw Data (7–35)	HIIT (*n* = 18)	25.89 ± 4.27	29.06 ± 3.81	10.9	−1.791	0.000 *
	CAT (*n* = 16)	24.69 ± 2.52	29.75 ± 3.02	17.0	−1.626	0.000 *
	CON (*n* = 14)	22.86 ± 3.57	21.14 ± 3.55	−8.1	0.425	0.114
Transformed score (4–20)	HIIT (*n* = 18)	14.83 ± 2.46	16.28 ± 2.32	8.9	−1.324	0.000 *
	CAT (*n* = 16)	13.94 ± 1.39	17.00 ± 1.71	18.0	−1.759	0.000 *
	CON (*n* = 14)	13.14 ± 2.11	11.93 ± 2.13	−10.1	0.498	0.069
Transformed score (0–100)	HIIT (*n* = 18)	67.61 ± 15.76	78.72 ± 13.65	14.1	−1.571	0.000 *
	CAT (*n* = 16)	62.26 ± 8.75	78.19 ± 15.13	20.3	−0.929	0.001 *
	CON (*n* = 14)	57.36 ± 13.08	48.29 ± 14.04	−18.7	0.541	0.051
Psychological State						
Raw Data (6–30)	HIIT (*n* = 18)	20.94 ± 2.84	24.22 ± 3.32	13.5	−1.955	0.000 *
	CAT (*n* = 16)	22.81 ± 1.76	25.44 ± 2.37	10.3	−1.007	0.001 *
	CON (*n* = 14)	19.79 ± 3.29	19.29 ± 4.50	−2.6	0.122	0.636
Transformed score (4–20)	HIIT (*n* = 18)	14.06 ± 1.86	16.28 ± 2.32	13.6	−1.860	0.000 *
	CAT (*n* = 16)	15.13 ± 1.26	17.00 ± 1.59	11.0	−1.067	0.000 *
	CON (*n* = 14)	13.07 ± 2.13	12.86 ± 2.93	−1.6	0.081	0.752
Transformed score (0–100)	HIIT (*n* = 18)	62.83 ± 11.63	75.89 ± 13.43	17.2	−1.894	0.000 *
	CAT (*n* = 16)	69.63 ± 7.83	81.50 ± 9.97	14.5	−1.089	0.000 *
	CON (*n* = 14)	56.64 ± 13.31	55.43 ± 18.16	−2.2	0.074	0.773
Social Relationships						
Raw Data (3–15)	HIIT (*n* = 18)	10.17 ± 2.18	11.83 ± 1.98	14.0	−0.455	0.001 *
	CAT (*n* = 16)	11.56 ± 1.46	12.25 ± 1.48	5.6	−0.367	0.143
	CON (*n* = 14)	10.14 ± 1.75	12.64 ± 2.76	19.7	0.194	0.453
Transformed score (4–20)	HIIT (*n* = 18)	13.22 ± 3.57	15.17 ± 4.18	12.9	−0.462	0.059
	CAT (*n* = 16)	15.44 ± 2.03	16.38 ± 2.06	5.7	−0.336	0.177
	CON (*n* = 14)	13.43 ± 2.44	12.64 ± 2.76	−6.3	0.189	0.465
Transformed score (0–100)	HIIT (*n* = 18)	57.61 ± 22.45	69.89 ± 25.92	16.1	−0.462	0.056
	CAT (*n* = 16)	71.50 ± 12.78	77.38 ± 12.96	7.5	−0.333	0.181
	CON (*n* = 14)	58.86 ± 15.35	53.93 ± 17.28	−9.1	0.189	0.467
Environmental Point						
Raw Data (8–40)	HIIT (*n* = 18)	25.17 ± 3.63	29.61 ± 4.35	15.0	−1.840	0.000 *
	CAT (*n* = 16)	28.13 ± 3.05	32.94 ± 4.22	14.6	−1.202	0.000 *
	CON (*n* = 14)	26.07 ± 4.36	45.00 ± 14.45	42.1	0.961	0.003 *
Transformed score (4–20)	HIIT (*n* = 18)	12.83 ± 1.82	15.06 ± 2.13	14.8	−1.820	0.000 *
	CAT (*n* = 16)	14.44 ± 1.55	16.63 ± 2.06	13.2	−1.180	0.000 *
	CON (*n* = 14)	13.29 ± 2.16	11.29 ± 2.13	−17.7	0.923	0.000 *
Transformed score (0–100)	HIIT (*n* = 18)	55.33 ± 11.33	29.61 ± 4.35	−86.9	−1.852	0.000 *
	CAT (*n* = 16)	65.38 ± 9.63	79.19 ± 12.94	17.4	−1.178	0.000 *
	CON (*n* = 14)	58.14 ± 13.42	45.00 ± 14.45	−29.2	0.926	0.003 *

Values are means ± SD; HIIT: high-intensity interval training; CAT: continuous aerobic training and CON: control group; *: significance difference of pre-test and post-test.

**Table 2 ijerph-20-00653-t002:** Resting heart rate, systolic BP—resting blood pressure, diastolic BP—resting blood pressure, rate pressure product and mean arterial pressure in college-aged smokers following 8 weeks of high-intensity interval training (HIIT), continuous aerobic training (CAT), and a control group (Con).

Variable	Group	Baseline	8-Weeks	%∆	Effect Size: Cohen’s d	*p*-Value
RHR (bpm)	HIIT (*n* = 18)	66.61 ± 9.15	63.11 ± 5.57	−5.5	0.667	0.009 *
	CAT (*n* = 16)	70.13 ± 4.10	64.13 ± 5.16	−9.4	1.424	0.000 *
	CON (*n* = 14)	74.14 ± 7.51	73.29 ± 4.39	−1.2	0.089	0.728
SBP–RBP (mmHg)	HIIT (*n* = 18)	106.17 ± 12.89	106.22 ± 10.06	0.0	−0.008	0.971
	CAT (*n* = 16)	112.88 ± 5.26	111.50 ± 7.54	−1.2	0.159	0.514
	CON (*n* = 14)	115.21 ± 5.18	115.57 ± 7.37	0.3	−0.048	0.851
DBP–RBP (mmHg)	HIIT (*n* = 18)	67.61 ± 7.95	68.33 ± 5.63	1.1	−0.147	0.523
	CAT (*n* = 16)	71.88 ± 4.35	69.25 ± 5.16	−3.8	0.421	0.096
	CON (*n* = 14)	69.93 ± 5.24	71.00 ± 4.49	0.1	−0.166	0.521
RPP (mmHg×bpm)	HIIT (*n* = 18)	73.36 ± 20.78	66.64 ± 10.93	−10.1	0.495	0.042 *
	CAT (*n* = 16)	79.28 ± 7.35	71.57 ± 8.12	−10.8	1.091	0.000 *
	CON (*n* = 14)	74.14 ± 7.51	83.15 ± 9.46	10.8	0.161	0.534
MAP (mmHg)	HIIT (*n* = 18)	80.80 ± 7.90	80.85 ± 6.56	0.1	−0.015	0.949
	CAT (*n* = 16)	85.54 ± 3.79	77.72 ± 20.52	−10.1	0.355	0.156
	CON (*n* = 14)	79.20 ± 22.64	85.86 ± 4.82	7.8	−0.268	0.306

Values are means ± SD; RHR: resting heart rate; BPM: beat per minute; SBP–RBP: systolic blood pressure—resting blood pressure; mmHg: millimetres of mercury; DBP–RBP: diastolic blood pressure—resting blood pressure; HIIT: high-intensity interval training; CAT: continuous aerobic training and CON: control group; BP: blood pressure; RPP: rate pressure product; MAP: mean arterial pressure; *: significance difference of pre-test and post-test.

**Table 3 ijerph-20-00653-t003:** Intra-comparisons of anthropometric variables (weight, height, body mass index, waist, hip, waist-hip ratio, sum of skinfolds, %body fat, fat mass, and lean mass) in college-aged smokers following 8 weeks of high-intensity interval training (HIIT), continuous aerobic training (CAT), and a control group (Con).

Variable	Group	Baseline	8-Weeks	%∆	Effect Size: Cohen’s d	*p*-Value
Weight (kg)	HIIT (*n* = 18)	66.69 ± 7.50	67.90 ± 7.61	1.8	−0.888	0.001 *
	CAT (*n* = 16)	73.79 ± 12.47	73.48 ± 10.97	−0.4	0.090	0.711
	CON (*n* = 14)	73.07 ± 5.06	70.77 ± 6.04	−3.2	0.983	0.002 *
BMI (kg/m^2^)	HIIT (*n* = 18)	22.56 ± 2.12	22.98 ± 2.10	1.8	−0.913	0.001 *
	CAT (*n* = 16)	23.37 ± 3.59	23.39 ± 3.590	0.1	−0.237	0.333
	CON (*n* = 14)	24.56 ± 2.32	23.65 ± 2.61	−3.7	1.008	0.000 *
Waist (cm)	HIIT (*n* = 18)	73.81 ± 3.67	74.02 ± 4.96	0.3	−0.077	0.737
	CAT (*n* = 16)	74.81 ± 8.57	73.48 ± 10.97	−1.8	−0.165	0.497
	CON (*n* = 14)	82.93 ± 9.41	84.92 ± 10.90	2.3	−0.293	0.265
Hip (cm)	HIIT (*n* = 18)	93.83 ± 5.06	94.34 ± 6.38	0.5	−0.087	0.703
	CAT (*n* = 16)	102.03 ± 10.50	98.95 ± 11.53	−3.1	0.331	0.183
	CON (*n* = 14)	100.93 ± 8.19	101.14 ± 11.44	0.2	−0.029	0.909
WHR	HIIT (*n* = 18)	0.79 ± 0.04	0.79 ± 0.04	0.0	0.141	0.540
	CAT (*n* = 16)	0.74 ± 0.09	0.76 ± 0.08	2.6	−0.388	0.123
	CON (*n* = 14)	0.82 ± 0.07	0.84 ± 0.10	2.3	−0.249	0.341
Sum of skinfolds (∑7)	HIIT (*n* = 18)	62.06 ± 17.85	73.61 ± 15.95	15.7	−1.233	0.000 *
	CAT (*n* = 16)	74.31 ± 13.97	88.56 ± 17.10	16.1	−2.040	0.000 *
	CON (*n* = 14)	69.07 ± 11.13	61.71 ± 9.36	−11.9	0.840	0.005 *
%BF	HIIT (*n* = 18)	8.06 ± 2.88	10.54 ± 3.20	23.5	−1.254	0.000 *
	CAT (*n* = 16)	10.03 ± 2.19	12.23 ± 2.58	18.0	−1.537	0.000 *
	CON (*n* = 14)	8.80 ± 1.86	8.79 ± 2.51	−0.1	0.003	0.991
Fat mass (kg)	HIIT (*n* = 18)	5.39 ± 2.11	7.19 ± 2.50	25.0	−1.237	0.000 *
	CAT (*n* = 16)	7.51 ± 2.62	9.15 ± 2.93	17.9	−1.386	0.000 *
	CON (*n* = 14)	6.43 ± 1.53	6.24 ± 2.03	−3.0	0.087	0.735
Lean mass (kg)	HIIT (*n* = 18)	61.28 ± 6.92	60.71 ± 6.74	−0.9	0.339	0.150
	CAT (*n* = 16)	66.25 ± 10.34	64.34 ± 8.53	−3.0	0.593	0.025 *
	CON (*n* = 14)	66.42 ± 4.67	64.52 ± 5.50	−2.9	0.642	0.024 *

Values are means ± SD; kg: kilograms; m: metre; cm: centimetre; %BF: body fat percentage; kg/m^2^: kilograms per metre squared; BMI: body mass index; WHR: waist–hip ratio; HIIT: high-intensity interval training; CAT: continuous aerobic training and CON: control group; *: significance difference of pre-test and post-test.

**Table 4 ijerph-20-00653-t004:** Intra-group comparisons of lung function variables (forced vital capacity, forced expiratory volume, forced vital capacity/forced expiratory volume in one second (%), peak expiratory flow, and forced expiratory flow) in college-aged smokers following 8 weeks of high-intensity interval training (HIIT), continuous aerobic training (CAT), and a control group (Con).

Variable	Group	Baseline	8-Weeks	%∆	Effect Size: Cohen’s d	*p*-Value
FVC(L)–Meas	HIIT (*n* = 18)	3.91 ± 0.49	4.36 ± 0.69	10.3	−0.569	0.022 *
	CAT (*n* = 16)	3.95 ± 0.45	4.26 ± 0.58	7.3	−0.434	0.087
	CON (*n* = 14)	3.78 ± 0.21	3.49 ± 0.56	−8.3	0.470	0.084
FVC(L)–Pred%	HIIT (*n* = 18)	84.48 ± 6.40	89.57 ± 13.08	5.7	−0.386	0.105
	CAT (*n* = 16)	81.79 ± 4.75	86.76 ± 11.49	5.7	−0.509	0.049 *
	CON (*n* = 14)	80.79 ± 6.90	79.41 ± 9.40	−1.4	0.119	0.644
FEV(L)–Meas	HIIT (*n* = 18)	3.46 ± 0.32	3.70 ± 1.02	6.5	−0.243	0.295
	CAT (*n* = 16)	3.56 ± 0.28	3.81 ± 0.35	6.6	−0.505	0.050 *
	CON (*n* = 14)	3.46 ± 0.10	3.42 ± 0.36	−1.2	0.100	0.698
FEV(L)–Pred%	HIIT (*n* = 18)	87.19 ± 4.80	93.78 ± 9.66	7.0	−0.657	0.010 *
	CAT (*n* = 16)	86.47 ± 3.20	89.51 ± 8.66	3.4	−0.383	0.127
	CON (*n* = 14)	86.87 ± 4.92	85.93 ± 9.30	−1.1	0.084	0.743
FVC/FEV_1_ (%)–Meas	HIIT (*n* = 18)	87.95 ± 5.06	89.34 ± 4.77	1.6	−0.195	0.397
	CAT (*n* = 16)	90.54 ± 3.64	91.32 ± 4.56	0.9	−0.154	0.527
	CON (*n* = 14)	91.83 ± 3.04	87.24 ± 4.53	−5.3	0.716	0.014 *
FVC/FEV_1_ (%)–Pred%	HIIT (*n* = 18)	106.36 ± 5.60	108.18 ± 5.87	1.7	−0.222	0.338
	CAT (*n* = 16)	109.24 ± 4.56	108.89 ± 4.04	−0.3	0.066	0.786
	CON (*n* = 14)	110.67 ± 3.51	104.84 ± 6.19	−0.1	0.670	0.020 *
PEF (L/s)–Meas	HIIT (*n* = 18)	8.11 ± 1.31	8.90 ± 1.21	8.9	−0.338	0.151
	CAT (*n* = 16)	8.36 ± 0.61	9.30 ± 1.28	10.1	−0.766	0.006 *
	CON (*n* = 14)	8.62 ± 0.58	7.06 ± 1.34	−22.1	0.966	0.002 *
PEF (L/s)–Pred%	HIIT (*n* = 18)	86.92 ± 13.51	92.23 ± 12.21	5.8	−0.228	0.325
	CAT (*n* = 16)	87.79 ± 8.04	97.04 ± 10.72	9.5	−0.697	0.010 *
	CON (*n* = 14)	91.89 ± 6.51	79.30 ± 12.79	−15.9	0.742	0.011 *
FEF_25 (L/s)–Meas	HIIT (*n* = 18)	6.93 ± 0.92	7.40 ± 0.56	0.1	−0.377	0.112
	CAT (*n* = 16)	7.08 ± 0.27	7. 07 ± 1.13	−1.1	−0.235	0.965
	CON (*n* = 14)	7.18 ± 0.29	6.70 ± 1.05	−7.2	0.371	0.164
FEF_25 (L/s)–Pred%	HIIT (*n* = 18)	87.49 ± 10.50	94.28 ± 9.93	7.2	−0.442	0.066
	CAT (*n* = 16)	88.28 ± 5.96	91.18 ± 13.09	3.18	−0.235	0.337
	CON (*n* = 14)	90.30 ± 4.56	84.82 ± 9.40	−6.5	0.460	0.091
FEF_50 (L/s)–Meas	HIIT (*n* = 18)	4.16 ± 0.61	4.60 ± 0.51	9.6	−0.511	0.037 *
	CAT (*n* = 16)	4.37 ± 0.25	4.79 ± 0.95	8.8	−0.423	0.095
	CON (*n* = 14)	4.37 ± 0.33	4.09 ± 0.46	−6.8	0.520	0.059
FEF_50 (L/s)–Pred%	HIIT (*n* = 18)	78.69 ± 10.21	85.22 ± 7.43	7.7	−0.512	0.036 *
	CAT (*n* = 16)	80.07 ± 5.11	76.63 ± 20.90	−4.5	0.152	0.531
	CON (*n* = 14)	84.00 ± 6.87	76.49 ± 7.06	−9.8	0.806	0.007 *
FEF_75 (L/s)–Meas	HIIT (*n* = 18)	1.63 ± 0.29	2.04 ± 0.50	20.1	−0.628	0.013 *
	CAT (*n* = 16)	1.80 ± 0.17	2.00 ± 0.65	10.0	−0.315	0.204
	CON (*n* = 14)	1.79 ± 0.16	1.65 ± 0.26	−8.5	0.453	0.095
FEF_75 (L/s)–Pred%	HIIT (*n* = 18)	68.23 ± 11.61	77.05 ± 12.07	11.4	−0.445	0.064
	CAT (*n* = 16)	70.83 ± 7.46	69.71 ± 7.68	−1.6	0.155	0.524
	CON (*n* = 14)	74.77 ± 7.44	66.74 ± 7.28	−12.0	0.769	0.009 *

Values are means ± SD; HIIT: high-intensity interval training; CAT: continuous aerobic training; CON: control group; FVC: forced vital capacity; FEV: forced expiratory volume; FVC/FEV_1_ (%): forced vital capacity/forced expiratory volume in one second (%); PEF: peak expiratory flow; FEF: forced expiratory flow; L/s: litres per second; Meas: measured; Pred%: predicted percentage; *: significance difference of pre-test and post-test.

**Table 5 ijerph-20-00653-t005:** Intra-group comparisons of cardiorespiratory endurance variables (steps/min, VO_2max_ and heart rate) in college-aged smokers following 8 weeks of high-intensity interval training (HIIT), continuous aerobic training (CAT), and a control group (Con).

Variable	Group	Baseline	8-Weeks	%∆	Effect Size: Cohen’s d	*p*-Value
Steps/min	HIIT (*n* = 18)	28.28 ± 3.79	29.67 ± 1.24	4.7	−0.365	0.123
	CAT (*n* = 16)	28.31 ± 1.58	29.06 ± 2.62	2.6	−0.299	0.227
	CON (*n* = 14)	24.64 ± 2.82	25.43 ± 2.90	3.1	−0.202	0.437
VO_2max_ (mL/kg/min)	HIIT (*n* = 18)	26.35 ± 3.06	27.47 ± 1.00	4.1	−0.365	0.123
	CAT (*n* = 16)	26.38 ± 1.28	26.98 ± 2.12	2.2	−0.300	0.226
	CON (*n* = 14)	23.41 ± 2.28	24.05 ± 2.34	2.7	−0.202	0.437
HR (BPM)	HIIT (*n* = 18)	75.22 ± 17.60	70.83 ± 10.67	−6.2	0.322	0.171
	CAT (*n* = 16)	98.25 ± 11.77	86.19 ± 12.17	−14.0	0.969	0.001 *
	CON (*n* = 14)	97.29 ± 16.06	102.07 ± 13.85	4.7	−0.246	0.346

Values are means ± SD; HIIT: high-intensity interval training; CAT: continuous aerobic training and CON: control group; VO_2max_: volume of maximum oxygen consumption; mL/kg/min: millilitre per kilogram per minute; HR: heart rate and BPM: beat per minute; *: significance difference of pre-test and post-test.

## Data Availability

Not applicable.

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
