# Peer review of "Effects of High-Intensity Interval Training and Continuous Aerobic Training on Health-Fitness, Health Related Quality of Life, and Psychological Measures in College-Aged Smokers"

_ijerph, 2022, doi:10.3390/ijerph20010653_

Round 1
Reviewer 1 Report
First of all, thank you for the opportunity to review this article. Below are some recommendations for authors.
Abstract
-I would recommend that you do not use abbreviations in the abstract, but only once the terms are mentioned in the main text.
Introduction
-The first line of the introduction should include a reference.
-Line 49-50 should be rewritten to facilitate understanding. I don't know if it is really well written, I think it repeats information.
-Line 56, The definition of HIIT needs to be rewritten as it is not true. HIIT target intensity during is usually “near maximal” or between 80 and 100% of maximal heart rate (HRmax) or maximum oxygen consumption (VO2max), while sprint interval training (SIT protocols usually involve “all-out” efforts. See "Gibala, M. J., Gillen, J. B., and Percival, M. E. (2014). Physiological and healthrelated adaptations to low-volume interval training: influences of nutrition and sex. Sport Med. 44, 127–137. doi: 10.1007/s40279-014-0259-6"
-Similarly, I would include in the introduction a paragraph stating the levels of smoking in the university population, its effects, as well as both preventive factors and those that favour tobacco consumption.
Methods
-The sub-section "Participants" should include the socio-demographic characteristics of the sample.
-The numbers of sub-sections "Lung Function Assessment", "Cardiorespiratory (CR) Endurance Assessment", "Intervention" and "Statistical analysis" are not correct.
-Beyond that, the "Methods" section is well written, structured and substantiated. Congratulations.
Results
-The results section is very well written and structured.
-However, I propose to include top and bottom borders in the tables for each of the variables, in order to facilitate the reader's understanding.
Discussion
-Line 298, the reference is not in the correct format.
-Line 313, the references are not in the correct format.
-I don't really understand how increasing weight and body fat while reducing muscle mass can be a positive effect for participants. Please elaborate on this point.
Conclusion
-The conclusions are well-written and concise
Author Response
Please see the attachment: Response to Reviewer 1 comments

Reviewer 2 Report
Effects of High-Intensity Interval Training and Continuous Aerobic Training on Health-Fitness, Health Related Quality of Life, and Psychological Measures in College-aged Smokers
First of all, the reviewer would like to thank the authors for their work and efforts in trying to improve sports science knowledge.
General comments to the authors
The article is investigating of exercise on health-fitness, health related quality of life (HRQOL) and psychological measures in college-age smokers. Overall, the study is well designed and well-written, with a great introduction proposing the usefulness of the topic and a clear outline of the research question. I suggest that the author modify/include some suggestions in order to improve the manuscript prior to be published:
Abstract
P 1 Line 12 and 16: It should be excluded (n=60)
Introduction section
P 2 Line 44: Is this [4; 5; 6; 7; 8) a right way to show references throughout the manuscript? Please check and fix them all references throughout the manuscript.
P 2 Line 52: I think that these articles about the effect of exercise on depressive indications and mental health should be added.
De Moor, M. H., Beem, A. L., Stubbe, J. H., Boomsma, D. I., & De Geus, E. J. (2006). Regular exercise, anxiety, depression and personality: a population-based study. Preventive medicine, 42(4), 273-279.
Arslan, E., Can, S., & Demirkan, E. (2017). Effect of short-term aerobic and combined training program on body composition, lipids profile and psychological health in premenopausal women. Science & Sports, 32(2), 106-113.
Nabkasorn, C., Miyai, N., Sootmongkol, A., Junprasert, S., Yamamoto, H., Arita, M., & Miyashita, K. (2006). Effects of physical exercise on depression, neuroendocrine stress hormones and physiological fitness in adolescent females with depressive symptoms. European journal of public health, 16(2), 179-184.
P 2 Line 64: I think that these articles about the effect of HIIT should be added to support their ideas..
Soylu, Y., Arslan, E., Sogut, M., Kilit, B., & Clemente, F. (2021). Effects of self-paced high-intensity interval training and moderateintensity continuous training on the physical performance and psychophysiological responses in recreationally active young adults. Biology of Sport, 38(4), 555-562.
Oliveira, B. R. R., Santos, T. M., Kilpatrick, M., Pires, F. O., & Deslandes, A. C. (2018). Affective and enjoyment responses in high intensity interval training and continuous training: A systematic review and meta-analysis. PloS one, 13(6), e0197124.
Methods section
Number of ethical file should be added.
Statistical Analysis
If you want, the authors should be added descriptors of Cohen’s d in Tables (for example small, medium and large)
Discussion section
Overall the discussion is well-written and incorporates relevant literature.
P 9 Line 281: I think that this sentence is very long. It should be separated into 2 sentences.
P 10 Line 340: I think that this sentence is very long. It should be separated into 2 sentences.
Tables and Figures
These sections are well designed and well-written. However the authors might be added a figure.
Author Response
Please see the attachement: Response to reviewer 2 comments
